# Agricultural Insurance and Agricultural Fertilizer Non-Point Source Pollution: Evidence from China's Policy-Based Agricultural Insurance Pilot

**Ziheng Niu [1], Feng Yi [2,*] and Chen Chen [3]**

[1]   Innovative Development Institute, Anhui University, Hefei 230039, China; nzh15006899282@163.com
[2]   School of Economics, Anhui University, Hefei 230601, China
[3]   School of Economics and Management, Beijing Forestry University, Beijing 100083, China; oshkkk94@163.com
**\***   Correspondence: yf13739224010@163.com

**Abstract:** For a long time, the relationship between agricultural insurance and the input of chemical fertilizer has been controversial. Since the pilot of policy-based agricultural insurance in China, most scholars have only paid attention to the role of the policy in ensuring farmers' income and reducing farmers' poverty, but its possible negative impact on the agricultural ecological environment is often ignored. If the pilot of this policy motivates farmers to apply more chemical fertilizers, which in turn causes more serious environmental problems, this would be contrary to the goals of the policy itself. Using the panel data of 31 provinces from 2000 to 2020 in China, this paper regards the pilot of policy-based agricultural insurance as a quasi-natural experiment and uses a difference-in-difference model to evaluate the impact of policy-based agricultural insurance on agricultural fertilizer non-point source pollution. The research results show that the pilot of policy-based agricultural insurance has aggravated the non-point source pollution of agricultural fertilizers in China. After a series of robustness tests, the research conclusion is still valid. At the same time, the effect of policy-based agricultural insurance aggravating agricultural fertilizer non-point source pollution had a lasting impact for 4 years during the pilot period and did not disappear until the policy-based agricultural insurance was fully covered. In addition, the heterogeneity results show that farmers in eastern China and high-disaster areas have a higher probability of moral hazard with overuse after purchasing policy-based agricultural insurance.

**Keywords:** policy-based agricultural insurance; agricultural fertilizer non-point source pollution; moral hazard; difference-in-difference model

## 1. Introduction

For a long time, the passive situation of farmers "relying on the weather for food" in China's agricultural production has not been fundamentally changed, and the ability of farmers to withstand natural disasters and other agricultural risks is still not strong [1]. In order to ensure the agricultural production safety of Chinese farmers and further enhance the ability of farmers to resist agricultural risks, in 2007, the Chinese government began to vigorously carry out the pilot work of policy-based agricultural insurance, and in 2012, policy-based agricultural insurance achieved full coverage. China's policy-based agricultural insurance is promoted by the government and has a non-profit nature, with subsidies jointly provided by central and local finance. Compared with general agricultural insurance, farmers' insurance costs are lower, and this plays a pivotal role in post-disaster compensation and weakening agricultural risks [2]. By 2015, the insured amount and signed premium of policy-based agricultural insurance in China accounted for more than 80% of all agricultural insurance. After more than 10 years of development, policy-based agricultural insurance has gradually become the main way of purchasing insurance for farmers in China.

Current studies on policy-based agricultural insurance mainly focus on evaluating the impact of policy-based agricultural insurance on helping farmers increase their income and reduce poverty [1–4]. Some studies have confirmed that policy-based agricultural insurance can change farmers' planting behaviors and enhance farmers' tendencies to specialize in planting [5]. However, few studies have focused on the impact of policy-based agricultural insurance on agricultural non-point source pollution.

China uses the largest amount of chemical fertilizer in the world, and the high input of chemical fertilizer makes agricultural non-point source pollution very prominent [6]. Existing studies have shown that the moral hazard problem (Farmer's moral hazard problem can be interpreted as farmers in the dishonest acts of cast or opportunistic behavior. Particularly under the protection of agricultural insurance, farmers tend to take the risk of production and business activity after being insured or reduce the agricultural production management level and even take the action of man-made destruction, increasing the range and extent of loss and thus increasing the probability of risk occurrence [7].) is widespread in agricultural insurance, and farmers' purchasing agricultural insurance will change their behavior in fertilizer input. Some scholars believe that chemical fertilizer is a kind of agricultural production input with high risk, and farmers tend to adopt agricultural production modes with higher risk and increased input of chemical fertilizer factors in order to obtain higher expected returns after purchasing insurance. For example, Horowitz and Lichtenber [8], based on a sample of 376 farms, found that the input of chemical fertilizer on farms increased after the farms purchased insurance. Zhong et al. [9] obtained similar research conclusions by using 340 cotton farmers' samples in China. Meanwhile, a recent study by He et al. [10] examined the impact of production cost crop insurance on farmers' fertilizer input by using the dataset of corn farmers in the Philippines, and its theoretical analysis showed that due to the existence of moral hazard, its impact on the fertilizer input may increase or decrease. Through empirical analysis, it was found that farmers purchasing production cost crop insurance increased their fertilizer input. Some scholars also believe that in order to obtain income compensation, farmers will neglect agricultural risk prevention and reduce the agricultural production inputs, which will reduce the input of chemical fertilizer. For example, Smith and Goodwin [11] used 235 farm samples in Kansas, USA and confirmed that farms purchasing insurance reduced the input of chemical fertilizer. Zhang et al. [12] also found that farmers reduced their fertilizer input after purchasing insurance by using 552 samples of vegetable farmers in China. At the same time, we also found that a small number of studies, even based on cross-sectional data obtained by the same survey institutions, had inconsistent conclusions. For example, Mishra et al. [13] replaced the amount of chemical fertilizer input with the consumption expenditure of chemical fertilizer input and studied a sample of 865 farmers obtained from the American Agricultural Resource Management Survey (AARMS) in 1998. They found that farmers' consumption expenditures on chemical fertilizer input decreased after purchasing insurance. However, Chang and Mishra [14] used the sample of 1757 farmers obtained from AARMS in 2003 and found that farmers increased their consumption expenditures on fertilizer input after purchasing insurance.

Up to now, China's policy-based agricultural insurance has been carried out for more than 10 years and experienced a great change from pilot to comprehensive coverage. What is the impact of China's policy-based agricultural insurance on China's agricultural fertilizer non-point source pollution? In the long run, if policy-based agricultural insurance stimulates farmers to apply more chemical fertilizer, which leads to more serious agricultural ecological environment problems, it will run counter to the policy's goal of policy-based agricultural insurance itself. Therefore, it is of great practical significance to evaluate the impact of policy-based agricultural insurance on agricultural fertilizer non-point source pollution. At the same time, as far as the above literature on the relationship between agricultural insurance and fertilizer input are concerned, they are all from the perspective of farmers. Due to different sample selection, their research conclusions may be inconsistent,

and their conclusions are not extrapolated enough. Therefore, it is necessary to conduct further reviews from the national perspective.

Based on the above analysis, using the inter-provincial panel data of 31 provinces in China from 2000 to 2020, this paper regarded the policy-based agricultural insurance pilot as a quasi-natural experiment and adopted the difference-in-difference (DID) model to evaluate the impact of policy-based agricultural insurance on agricultural fertilizer non-point source pollution. This paper aims to evaluate the policy value of policy-based agricultural insurance pilot from the perspective of the agricultural environment in order to expect that the research conclusions of this paper can provide a relevant support basis for continuing to optimize the policy content. Compared with previous studies, the possible innovations of this paper are as follows. In terms of the research perspective, this paper takes the pilot of China's policy-based agricultural insurance as an example to explore the relationship between agricultural insurance and agricultural fertilizer non-point source pollution from a macro perspective, which can effectively overcome the conclusion alienation caused by sample selection differences in previous micro studies and obtain more general research conclusions. In the identification strategy, this paper uses the DID model, which can effectively alleviate the potential endogenous problems of the pilot of policy-based agricultural insurance and obtain more reliable research conclusions. In terms of research significance, this paper evaluates the environmental effects of policy agricultural insurance pilot promotion, which provides a supporting basis for optimizing the policy content.

The other structure of this paper is as follows. The Section 2 covers mechanism analysis and disagreement. The Section 3 describes the model and data. The Section 4 gives the results and analysis. The Section 5 discusses the results. The Section 6 concludes the paper.

## 2. Mechanism Analysis and Disagreement

China's policy-based agricultural insurance is based on the market-oriented operation of insurance companies and supported by the government through premium subsidies and other policies to provide direct materialized cost insurance for the economic losses caused by natural disasters and accidents in the agricultural sector. It has the inclusive characteristics of "low premium, wide coverage and high guarantee". It is the main form of insurance for farmers in China. Moral hazard in agricultural insurance will cause farmers to change their fertilizer input behavior. However, academia has not reached a consensus on how to change farmers' fertilizer input behavior by participating in agricultural insurance [8–14]. According to the existing literature, the root cause of the divergence of views lies in the uncertainty between moral hazard with overuse and moral hazard with nonfeasance after farmers participating in the insurance scheme.

### 2.1. The Occurrence Mechanism of Moral Hazard with Overuse

The occurrence of moral hazard with overuse is one of the manifestations of the problem of moral hazard in agricultural insurance. The occurrence of moral hazard with overuse will prompt farmers to take more risky agricultural production actions [7]. Chemical fertilizer is often considered a high-risk agricultural production factor [8]. When increasing the input of chemical fertilizer, such a high-risk agricultural production factor not only increases the expected income of farmers but also increases the volatility of farmers' income and increases the risk of production reduction [15]. After farmers purchase policy-based agricultural insurance, under the protection of policy-based agricultural insurance, their agricultural risks are greatly reduced [9]. In order to obtain higher expected returns, farmers tend to take agricultural production actions with high risks and increase fertilizer input [14]. In other words, under the condition of the farmers purchasing policy-based agricultural insurance, even if farmers apply excessive chemical fertilizer in the pursuit of higher expected returns and increase the risk of yield reduction, they know that policy-based agricultural insurance will provide corresponding income compensation once the yield reduction occurs, so they will not reduce the input of chemical fertilizer due to the fear of increasing the risk of yield reduction.

In conclusion, the occurrence mechanism of moral hazard with overuse will reduce farmers' expectations of the risk of yield reduction caused by excessive fertilization and further encourage farmers to increase their input of chemical fertilizer in the pursuit of higher expected returns. With the excessive use of chemical fertilizer, the problem of agricultural fertilizer non-point source pollution will become more serious. As mentioned above, many scholars have confirmed the incentive effect of agricultural insurance on chemical fertilizer investment [8–10]. At the same time, the occurrence mechanism of moral hazard with overuse in agricultural insurance will not only change farmers' fertilizer inputs. Because the agricultural production property of pesticides is similar to that of chemical fertilizer, many studies also found that farmers' participation in agricultural insurance can encourage farmers to overuse pesticides. For example, the research of Hill et al. [16] found that farmers will increase their use of pesticides in order to obtain higher expected returns in Bangladesh. The occurrence mechanism of moral hazard with overuse is an important potential reason for agricultural insurance to aggravate the non-point source pollution of agricultural chemical fertilizer.

### 2.2. The Occurrence Mechanism of Moral Hazard with Nonfeasance

Smith and Goodwin [11] refuted the occurrence mechanism of moral hazard with overuse and believed that when the input of high-risk agricultural production factors such as chemical fertilizer is increased, the increase in income volatility will increase the possibility of compensation, but the increase in expected income will decrease the possibility of compensation. Therefore, if the increase in the fluctuation of farmers' income cannot offset the increase in their expected income, then after purchasing agricultural insurance, the possibility of farmers receiving income compensation due to the increase in fertilizer input will decrease, which will be unfavorable for farmers to increase their investment in fertilizers. At the same time, Smith and Goodwin [11] further pointed out that due to the occurrence of moral hazard with nonfeasance, farmers will neglect the prevention of agricultural risks, thereby reducing the input of fertilizer elements.

Specifically, when farmers purchase policy-based agricultural insurance, under the compensation mechanism of policy-based agricultural insurance, farmers' agricultural production actions will become more negative or conservative in order to obtain the expected income compensation, which is manifested in reducing the level of agricultural production management in agricultural production activities, actively slackening agricultural risk prevention, resulting in reduced fertilizer input [11,12].

In conclusion, the occurrence mechanism of moral hazard with nonfeasance will reduce farmers' enthusiasm for agricultural production, further promote farmers to improve the possibility of negative production in order to obtain income compensation and then reduce the input of agricultural production, which will also reduce the input of chemical fertilizer. As mentioned above, many scholars have confirmed the negative effect of agricultural insurance on chemical fertilizer input [11–13]. At the same time, not only the input of chemical fertilizer but also the occurrence of moral hazard with nonfeasance will make farmers' production behaviors more slacked, which will also reduce farmers' enthusiasm for pesticide application. For example, Han et al. [17] found that farmers who participate in agricultural insurance will actively reduce the use intensity of pesticides in order to obtain income compensation in China. Although the occurrence mechanism of moral hazard with nonfeasance is not enough to cause agricultural fertilizer non-point source pollution, it reduces farmers' production efficiency and enthusiasm and is also detrimental to the development of agriculture in the long run.

### 2.3. Research Framework

Based on the above analysis, this paper establishes a theoretical analysis framework for the impact of policy-based agricultural insurance on farmers' fertilizer non-point source pollution from the perspective of moral hazard as shown in Figure 1. The existence of moral hazard is not conducive to the healthy development of agricultural insurance, but it

is always rooted in the "soil" of agricultural insurance. It is essentially a kind of behavior driven by interests but in line with "rationality" [7]. Then what is the impact of China's policy-based agricultural insurance on agricultural fertilizer non-point source pollution? This has yet to be validated using national data.

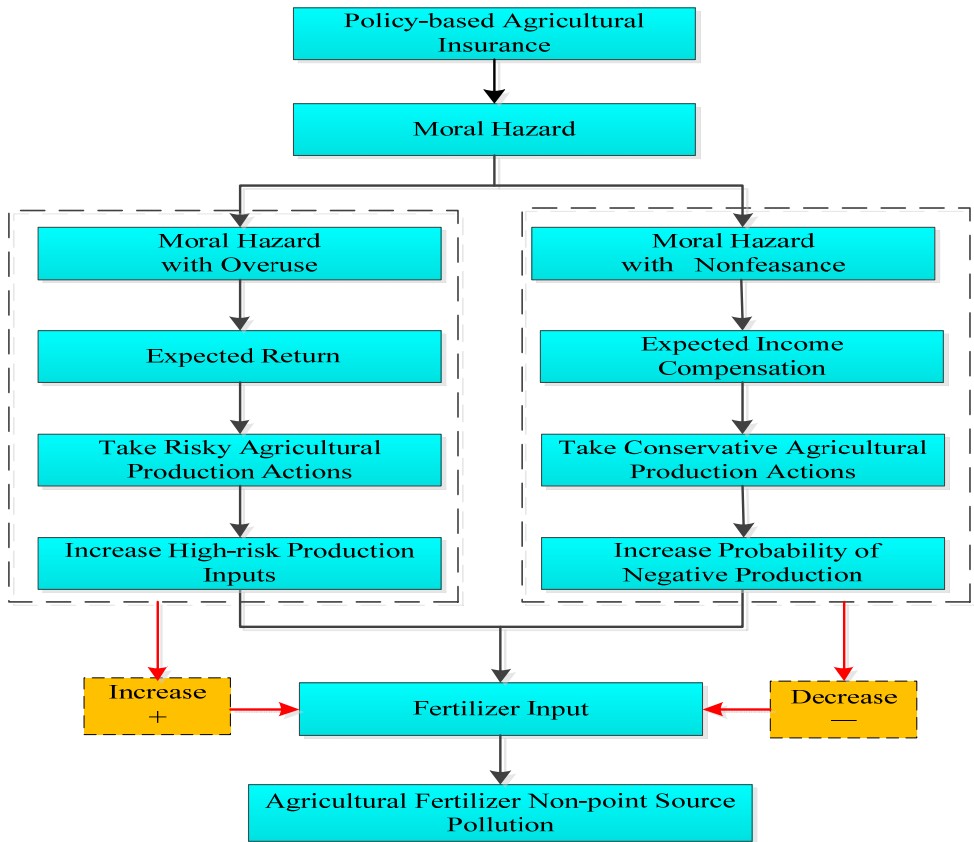

**Figure 1.** Research framework. Source: collating of the author.

## 3. Model and Data Description

### 3.1. Model

Since the pilot work of policy-based agricultural insurance is carried out by gradually increasing the number of pilot provinces according to the time, and the policy implementation time of different provinces is not consistent, it is necessary to establish a time-varying DID model to evaluate the impact of policy-based agricultural insurance on agricultural fertilizer non-point source pollution. Based on the research of Fu and Liang [5], the following DID model was constructed in this paper:

$$TE_{i,t} = \beta_0 + \theta Treat_i \cdot Time_t + \delta Control_{i,t} + \mu_i + \lambda_t + \varepsilon_{i,t} \tag{1}$$

In the above formula, *TE* is the explained variable and represents the pollution intensity of the agricultural fertilizer non-point source. *Treat* is the dummy variable of the policy area, where the value is 1 when the area is the pilot province of policy-based agricultural insurance, and otherwise, the value is 0. *Time* is the dummy variable of policy time, where the value of the year in which the policy of agricultural insurance is implemented in the province is 1, and otherwise, it is 0. *Treat·Time* is the core explanatory variable of the model, which represents the policy variable of the policy-based agricultural insurance pilot. $\theta$ is its estimated coefficient, which represents the actual policy effect of policy-based agricultural insurance on agricultural fertilizer non-point source pollution. *Control* is a series of control variables, including the sown area of crops, agricultural mechanization level, urbanization level, disaster degree, per capita income level of rural residents and its square term, while $\delta$

is its estimated coefficient vector. $\beta_0$ is a constant term, $\mu_i$ is the fixed effect of the province, $\lambda_t$ is the fixed effect of time, $\varepsilon$ is a random error term and $i$ and $t$ represent the province and year, respectively.

The main limitation of using the DID model is ensuring that the policy treatment group and the control group are comparable before the policy is implemented; that is, the treatment group and the control group should have a parallel trend. In other words, it is necessary to test that the variation trend of the intensity of agricultural fertilizer non-point source pollution over time between the treatment group and the control group is consistent before the pilot promotion of policy-based agricultural insurance is carried out. Based on the study of Beck et al. [18], this paper adopts the event study method to investigate whether the parallel trend test is met and sets the following model:

$$TE_{i,t} = \beta_0 + \sum_{d=-7,d\neq0}^{d=10} \theta_d Treat_i \cdot Time_d + \delta Control_{i,t} + \mu_i + \lambda_t + \varepsilon_{i,t} \tag{2}$$

In the above formula, $Time_d$ is the dummy variable of year $d$ before and after the pilot promotion of policy-based agricultural insurance, and the value range is $-7 \leq d \leq 7$. If d = $-7$, the value is 1 when the year is 2000; otherwise, the value is 0. In addition, by setting $d \neq 0$, this paper takes the year of the pilot promotion of policy-based agricultural insurance as the control. According to the study of Beck et al. [18], if the magnitude of the estimated coefficient $\theta_d$ fluctuates at 0 before the policy is implemented and cannot pass the significance test but after the policy is implemented, the estimated coefficient $\theta_d$ deviates significantly from 0 and starts to pass the significance test, and then the parallel trend can be satisfied.

Due to the obvious systematic regional differences and disaster levels in different provinces in China, there may be differences in the impact of a policy-based agricultural insurance pilot on agricultural fertilizer non-point source pollution in provinces with different locations and different disaster degrees. Therefore, this paper further investigated the heterogeneous impact of policy-based agricultural insurance on agricultural fertilizer non-point source pollution from regional differences and differences in disaster degrees. On this basis, this paper establishes the following the difference-in-difference-in-difference (DDD) model by referring to the study of Zhou et al. [19]:

$$\begin{aligned} TE_{i,t} = \beta_0 &+ \theta_1 Treat_i \cdot Time_t \cdot Type_i + \theta Treat_i \cdot Time_t + \alpha_1 Treat_i \cdot Type_i + \alpha_2 Time_t \cdot Type_i \\ &+ \delta Control_{i,t} + \mu_i + \lambda_t + \varepsilon_{i,t} \end{aligned} \tag{3}$$

In the above formula, $Type$ is the dummy variable of the province type. Specifically, due to the large systematic difference between eastern, central and western in China, this paper set regional dummy variable based on the study of Wang et al. [20]. When a province is located in the eastern part, $Type$ is assigned to be 1. When the province is located in the midwestern part, $Type$ is assigned to be 0. At the same time, this paper draws lessons from Liang [21] on the classification of disaster degree for all provinces in China. When a province is located in a high-risk area, $Type$ is assigned to be 1. When the province is in a low-risk area, $Type$ is assigned to be 0. $\theta_1$ is the estimation coefficient of interest in this paper, representing the heterogeneous influence of policy-based agricultural insurance on agricultural fertilizer non-point source pollution.

### 3.2. Variable Selection

#### 3.2.1. Dependent Variable

The dependent variable in this paper is the pollution intensity of the agricultural fertilizer non-point source. Agricultural fertilizer non-point source pollution is characterized by fast diffusion and difficult data statistics [22]. The existing studies generally adopt the unit investigation and evaluation method for measurement [23,24]. Therefore, based on the study of Shi et al. [25], nitrogen fertilizer, phosphorus fertilizer and compound fertilizer were identified as the investigation unit of agricultural fertilizer non-point source pollution. According to the chemical composition proportion of nitrogen and phosphorus in nitrogen

fertilizer, phosphorus fertilizer and compound fertilizer, the pollutant-producing coefficient of the nitrogen element in nitrogen fertilizer, phosphorus fertilizer and compound fertilizer was determined to be 1, 0 and 0.33, respectively. The pollutant-producing coefficients of phosphorus were 0, 0.44 and 0.15, respectively. On this basis, referring to the fertilizer-losing coefficient summarized by Lai [26], the total emissions of agricultural fertilizer non-point source pollution are as follows:

$$E_{i,t} = \sum_{s=1}^{3} EU_{i,t,s} \cdot \kappa_s \cdot \nu_s \tag{4}$$

In the above formula, $E$ is the total amount of non-point source pollution of agricultural chemical fertilizer, $s \in [1, 2, 3]$ represents three different kinds of chemical fertilizer, $EU$ is the discounted application amount of the $s$ type of chemical fertilizer, $\kappa_s$ is the pollutant-producing coefficient of the $s$ type of chemical fertilizer, and $\nu_s$ is the fertilizer-losing coefficient of the $s$ type of chemical fertilizer. Furthermore, the agricultural fertilizer non-point source pollution intensity can be obtained as follows:

$$TE_{i,t} = E_{i,t}/S_{i,t} \tag{5}$$

In the above formula, $S_{i,t}$ is the total area sown for crops and $i$ and $t$ represent the province and year, respectively.

### 3.2.2. Core Independent Variable

The core independent variable of this paper is the policy variable of policy-based agricultural insurance. The pilot work of China's policy-based agricultural insurance started in 2007. Afterward, the pilot province increased year by year, and by 2012, China's policy-based agricultural insurance pilot work covered all provinces (except Hong Kong, Macao and Taiwan). Therefore, in this paper, all provinces are pilot provinces of policy-based agricultural insurance. However, for each policy pilot province, its policy pilot time is not consistent. This paper summarizes the promotion process of China's policy-based agricultural insurance pilot through the notices of relevant provinces and cities for policy-based agricultural insurance from 2007 to 2012 as shown in Table 1.

**Table 1.** China's policy-based agricultural insurance pilot process.

| Year | Pilot Provinces |
|------|-----------------|
| 2007 | Jilin, Inner Mongolia, Jiangsu, Hunan, Sichuan, Xinjiang |
| 2008 | Hebei, Liaoning, Heilongjiang, Anhui, Shandong, Henan, Hubei, Zhejiang, Fujian, Hainan |
| 2009 | Jiangxi |
| 2010 | Shanxi, Guangdong, Yunnan, Gansu, Qinghai, Ningxia |
| 2011 | Guangxi, Guizhou, Tibet, Shaanxi, Chongqing |
| 2012 | Beijing, Shanghai, Tianjin |

Source: collating of the author.

### 3.2.3. Control Variables

Based on existing studies, this paper selected the sown area of the crops (*Size*) [27], agricultural mechanization level (*Am*) [22], urbanization level (*Urban*) [28], disaster degree (*Ad*) [29], income level of rural residents (*Income*) and its square term (*Income*$^2$) [30] as the control variables affecting the pollution intensity of an agricultural fertilizer non-point source. *Size* was taken as the total sown area of the crops. *Am* was measured by the total power of agricultural machinery. *Ad* was determined by the proportion of the disaster formation area in the total disaster area. *Urban* was expressed by the proportion of the urban population in the total population. *Income* and *Income*$^2$ were calculated with the per capita net income of rural residents and its square term. In order to eliminate the influence of price factors, this paper used the year 2000 as the base period and adopted the consumer price index to carry out the smoothing treatment on the per capita income of rural residents. Meanwhile, since the National Bureau of Statistics reformed the survey method of income

and expenditure of rural and urban residents in 2013, the statistical aperture of its rural income accounting changed. Therefore, this paper used the per capita disposable income of rural residents as an alternative [31].

### 3.3. Data Sources and Descriptive Statistics

This paper used the panel data of 31 provinces (excluding Hong Kong, Macao and Taiwan) from 2000 to 2020 for empirical analysis in China. The data came from the *China Statistical Yearbook*, *China Agricultural Yearbook* and *China Rural Statistical Yearbook*. In addition, the statistical yearbook of each province served to supplement the missing values. The descriptive statistics of the variables are shown in Table 2.

**Table 2.** Descriptive statistics of the variables.

| Variables | N | Mean | Std. | Min | Max |
|:---:|:---:|:---:|:---:|:---:|:---:|
| *TE* | 651 | 0.428 | 0.298 | 0.063 | 1.475 |
| *Size* | 651 | 5.146 | 3.731 | 0.089 | 14.910 |
| *Am* | 651 | 2.763 | 2.687 | 0.094 | 13.350 |
| *Urban* | 651 | 0.508 | 0.156 | 0.189 | 0.942 |
| *Ad* | 651 | 0.504 | 0.152 | 0.000 | 0.913 |
| *Income* | 651 | 0.546 | 0.333 | 0.133 | 1.993 |
| *Income*$^2$ | 651 | 0.409 | 0.534 | 0.018 | 3.972 |

Source: collating of the author.

## 4. Results and Analysis

### 4.1. Estimation Results of the DID Model

The estimation results of the DID model are shown in Table 3, and the control variables in the regression models of Equations (1)–(4) are gradually added. In the regression models, with the increase in the control variables, the estimation coefficients of policy-based agricultural insurance were all significantly positive, which indicates that China's policy-based agricultural insurance has indeed aggravated the non-point source pollution of agricultural fertilizers. This means that policy-based agricultural insurance is more likely to trigger the mechanism of moral hazard with overuse. After farmers buy policy-based agricultural insurance, the agricultural risks they face are greatly reduced. Therefore, in order to pursue higher incomes, farmers adopt more risky agricultural production actions and then increase the input of chemical fertilizer, which aggravates the non-point source pollution of agricultural chemical fertilizer.

The research conclusion of this paper confirms the widespread phenomenon of moral hazard with overuse in China from a macro level. As the research of He et al. [10] showed, the impact of agricultural insurance on agricultural fertilizer input has two sides, and ultimately, whether to increase or reduce chemical fertilizer input depends on the intensity of the two types of moral hazard. In terms of the research findings of China's policy-based agricultural insurance, policy-based agricultural insurance has led to greater moral hazard with overuse, which has increased farmers' fertilizer input and exacerbated agricultural fertilizer non-point source pollution. This finding can be supported by the research conclusions of Horowitz and Lichtenber [8] and Chang and Mishra [14].

According to the estimation results of the regression model in Equation (4), the impact of *Am* on the intensity of agricultural fertilizer non-point source pollution was −0.009, which passed the significance test of 10%, indicating that the level of agricultural mechanization inhibited agricultural fertilizer non-point source pollution in China. The reason for this may be that with the continuous improvement of the level of agricultural mechanization in China, on the one hand, it promotes the refinement of farmers' agricultural production and then reduces the amount of fertilizer input. On the other hand, the spread of environmentally friendly technologies such as straw returning in agricultural production has been accelerated, which has a certain crowding out effect on fertilizer application, and the pollution degree of the fertilizer non-point source is reduced. The impact of *Size* on the

intensity of agricultural fertilizer non-point source pollution was −0.026, which passed the significance test of 1%, indicating that the increase in the crop sown area reduced the intensity of agricultural fertilizer non-point source pollution. The reason for this is that with the increase in the crop sown area, the agricultural fertilizer non-point source pollution is distributed to a certain extent. *Income* and *Income*$^2$ had significant negative and positive impacts on the intensity of agricultural fertilizer non-point source pollution, respectively, which confirms the hypothesis of the environmental Kuznets curve (EKC), which is consistent with the general conclusion of the existing literature; that is, there is an "inverted U-shaped" non-linear relationship between the per capita income of rural residents and agricultural fertilizer non-point source pollution in China [32]. In addition, *Urban* and *Ad* had no significant impact on agricultural fertilizer non-point source pollution.

**Table 3.** Estimation results of the DID model.

| Variables | (1) TE | (2) TE | (3) TE | (4) TE |
|---|---|---|---|---|
| *Treat·Time* | 0.028 * | 0.024 * | 0.024 * | 0.029 ** |
| | (0.015) | (0.015) | (0.015) | (0.015) |
| *Size* | | −0.022 *** | −0.023 *** | −0.026 *** |
| | | (0.006) | (0.006) | (0.006) |
| *Am* | | −0.005 | −0.005 | −0.009 * |
| | | (0.005) | (0.005) | (0.005) |
| *Urban* | | | −0.007 | −0.088 |
| | | | (0.069) | (0.071) |
| *Ad* | | | 0.026 | 0.019 |
| | | | (0.022) | (0.022) |
| *Income* | | | | 0.267 ** |
| | | | | (0.118) |
| *Income*$^2$ | | | | −0.129 *** |
| | | | | (0.040) |
| *Cons_* | 0.360 *** | 0.481 *** | 0.469 *** | 0.476 *** |
| | (0.013) | (0.032) | (0.046) | (0.051) |
| *Province Fixed* | Yes | Yes | Yes | Yes |
| *Year Fixed* | Yes | Yes | Yes | Yes |
| *N* | 651 | 651 | 651 | 651 |
| $R^2$ | 0.219 | 0.245 | 0.247 | 0.268 |

Note: ***, ** and * are significant at the level of 1%, 5% and 10%, respectively, and the numbers in brackets are standard errors. Source: collating of the author according to Stata17. The same applies to the table below.

## 4.2. Parallel Trend Test and Dynamic Influence Effect

The results of the parallel trend test using Equation (2) are shown in Figure 2. In the figure below, the vertical axis represents the estimated coefficient *θ*, the horizontal axis represents year *d* before and after the pilot promotion of policy-based agricultural insurance, and the solid line above and below the circle represents the 90% confidence interval. From the graphic results, it can be found that prior to the implementation of the policy, the estimated coefficient *θ* was less than zero and wandered toward zero, but not through the test of significance. After the policy implementation, the size of the estimated coefficient *θ* obviously deviated to zero, and there was a significantly positive influence, showing that the DID model met a parallel trend. At the same time, since 2008, the impact of the pilot promotion of policy-based agricultural insurance on the agricultural fertilizer non-point source pollution increased significantly and did not disappear until 2011, which means that during the pilot promotion period of policy-based agricultural insurance in China, the aggravating effect of policy-based agricultural insurance on the agricultural fertilizer non-point source pollution had a continuous impact for a period of 4 years. After 2011, due to the full coverage of the promotion of policy-based agricultural insurance, the effect was no longer significant.

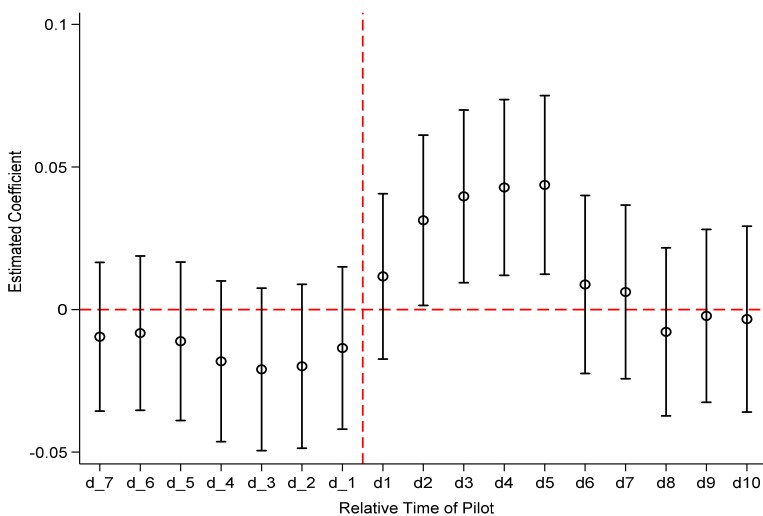

**Figure 2.** Parallel trend test.

*4.3. Robustness Test*

In order to verify the robustness of the above estimated results and improve the credibility of the estimated results, this paper carried out robustness tests from the following three aspects.

4.3.1. A Counterfactual Test of the Point at Which Policy Changes Occurred

In order to exclude the interference of the above estimated results by other policies or random factors, this paper used the study of Wang and Zhu [33] for reference and conducted a counterfactual test on the estimated results by changing the time point at which the policies occurred. To be specific, this paper assumed that the policy occurrence time of all pilot provinces was advanced by 1 year, 2 years and 3 years, and the DID model was reconstructed. The estimated results are shown in Table 4.

**Table 4.** Counterfactual test.

| Variables | 2004 TE | 2005 TE | 2006 TE |
|---|---|---|---|
| Treat·Time | −0.011 (0.025) | 0.008 (0.018) | 0.013 (0.011) |
| Control | Yes | Yes | Yes |
| Con_ | 0.568 *** (0.057) | 0.489 *** (0.061) | 0.413 *** (0.052) |
| Province Fixed | Yes | Yes | Yes |
| Year Fixed | Yes | Yes | Yes |
| N | 651 | 651 | 651 |
| $R^2$ | 0.244 | 0.249 | 0.251 |

Note: ***, ** and * are significant at the level of 1%, 5% and 10%, respectively.

The estimation results show that when the pilot promotion time of policy-based agricultural insurance in each pilot province was advanced by 1 year, 2 years and 3 years, the estimated coefficient of policy-based agricultural insurance was no longer significant. This means that the pilot promotion of policy-based agricultural insurance, rather than other agricultural support policies, is likely to aggravate agricultural fertilizer non-point source pollution.

4.3.2. Propensity Score-Matching Difference-in-Difference Model

In order to control the "selectivity bias" of the policy pilot, this paper further adopted a propensity score-matching difference-in-difference (PSM-DID) model to re-evaluate the

impact of a policy-based agricultural insurance pilot on agricultural fertilizer non-point source pollution. It is worth pointing out that, compared with the DID model, although the PSM-DID model could control the "selectivity bias" of the policy pilot, it often lost more of the sample size while requiring a larger common support domain.

The PSM-DID model should meet the balance test; that is, there is no significant difference in the characteristic variables after sample matching between the treatment group and the control group. In this paper, all the control variables, including the sown area of the crops, the level of agricultural mechanization, the level of urbanization, the degree of disaster, the per capita income level of rural residents and its square term, were required, and there was no significant difference between the treatment group and the control group after matching. The results of the balance test in this paper are shown in Table 5.

**Table 5.** Balance test.

| Variables | Sample Matching | Mean | | T-Value | *p*-Value |
|---|---|---|---|---|---|
| | | Treatment Group | Control Group | | |
| *Size* | Before | 5.563 | 4.599 | 3.290 *** | 0.001 |
| | After | 4.416 | 4.071 | 0.470 | 0.640 |
| *Am* | Before | 3.409 | 1.919 | 7.290 *** | 0.000 |
| | After | 2.387 | 2.096 | 0.600 | 0.553 |
| *Urban* | Before | 0.556 | 0.444 | 9.610 *** | 0.000 |
| | After | 0.533 | 0.542 | −0.210 | 0.831 |
| *Ad* | Before | 0.483 | 0.530 | −3.860 *** | 0.001 |
| | After | 0.512 | 0.476 | 1.220 | 0.227 |
| *Income* | Before | 0.712 | 0.329 | 17.760 *** | 0.000 |
| | After | 0.525 | 0.565 | −0.610 | 0.546 |
| *Income*$^2$ | Before | 0.609 | 0.148 | 12.090 *** | 0.000 |
| | After | 0.450 | 0.354 | 0.950 | 0.343 |

Note: ***, ** and * are significant at the level of 1%, 5% and 10%, respectively.

According to the balance test results, there were significant differences between the treatment group and the control group in the *Size*, *Am*, *Urban*, *Ad*, *Income* and *Income*$^2$ of samples before matching. After matching, there were no significant differences between the treatment group and the control group. This indicates that overall, the PSM-DID model passed the balance test.

On the basis of the PSM-DID model passing the balance test, the estimation results of the PSM-DID model are shown in Table 6. From the estimation results of the PSM-DID model, in models (1) and (2) below, the estimation coefficients of policy-based agricultural insurance were significantly positive, which again confirms the research conclusion that the pilot promotion of policy-based agricultural insurance aggravates agricultural fertilizer non-point source pollution in China. In addition, the estimation coefficient of policy-based agricultural insurance was in good agreement with the above estimated results in terms of its significance and direction, indicating that the above estimated results were robust.

**Table 6.** PSM-DID model.

| Variables | (1) TE | (2) TE |
|---|---|---|
| Treat·Time | 0.023 * | 0.026 ** |
| | (0.013) | (0.012) |
| Control | No | Yes |
| con_ | 0.541 *** | 0.608 *** |
| | (0.036) | (0.072) |
| Province Fixed | Yes | Yes |
| Year Fixed | Yes | Yes |
| N | 513 | 513 |
| $R^2$ | 0.279 | 0.285 |

Note: ***, ** and * are significant at the level of 1%, 5% and 10%, respectively.

### 4.3.3. Permutation Test

In order to further illustrate the validity of the estimation results of the DID model (i.e., the aggravation of agricultural fertilizer non-point source pollution is caused by the pilot promotion of policy-based agricultural insurance rather than other non-observational factors), this paper referred to the study of Lu et al. [34]. This was confirmed by a permutation test in the randomized change treatment group. The specific ideas of the method are as follows.

A new treatment group was randomly selected from 31 provinces and repeated 1000 times to obtain the 1000 coefficients of estimation. If the estimation coefficients of the actual treatment groups were significantly different from those of the randomly selected treatment groups, the robustness of the estimated results of the DID model could be proven. In other words, the aggravation of agricultural fertilizer non-point source pollution was indeed caused by the pilot promotion of policy-based agricultural insurance rather than other non-observable factors.

The results of the permutation test are shown in Figure 3, where the curve is the kernel density of the estimated coefficients and the dotted line is the estimation coefficient of actual policy-based agricultural insurance. It can be found that the actual coefficient was significantly different from the estimated coefficients obtained by randomly selecting treatment groups. Thus, it was confirmed that the causal effect of policy-based agricultural insurance aggravating agricultural fertilizer non-point source pollution was not caused by other unobserved factors. In other words, the DID model was robust.

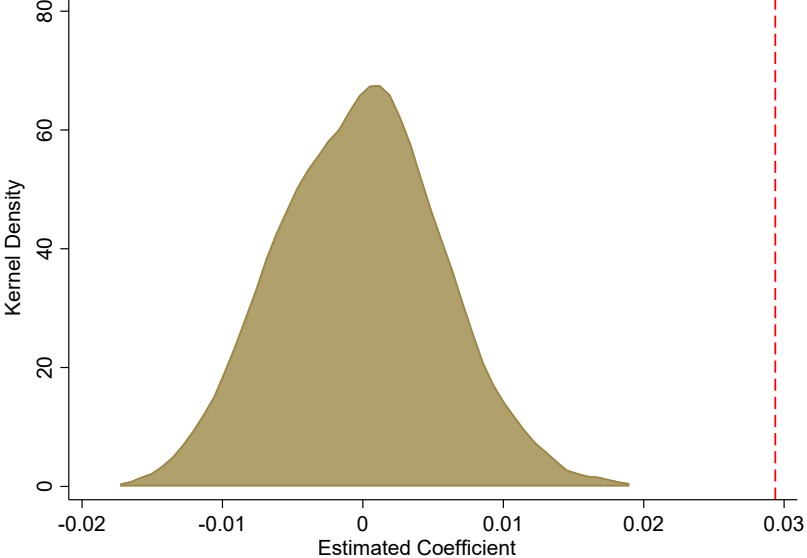

**Figure 3.** Permutation test.

#### 4.4. Heterogeneity Analysis

The estimation results of regional heterogeneity using Equation (3) are shown in models (1) and (2) of Table 7. As can be seen from the estimated results, $\theta_1$ was significantly positive regardless of the addition of control variables, which means that compared with the provinces located in the midwestern part, the policy-based agricultural insurance pilot aggravated agricultural fertilizer non-point source pollution to a greater extent in the eastern part. This may be because on the whole, farmers in eastern provinces are more prone to moral hazard with overuse after buying insurance. The estimation results of the heterogeneity of the disaster degree using Equation (3) is shown in models (3) and (4) in Table 7. As can be seen from the estimated results, $\theta_1$ was significantly positive regardless of whether the control variables were added, which indicates that policy-based agricultural insurance aggravated agricultural fertilizer non-point source pollution to a greater extent in areas with higher disaster degrees. The above results indicate that, compared with areas with low disaster degrees, farmers in areas with high disaster degrees were more likely to have moral hazard with overuse after purchasing policy-based agricultural insurance.

**Table 7.** Heterogeneity results.

| Variables | Heterogeneity (Location) | | Heterogeneity (Disaster Degree) | |
|---|---|---|---|---|
| | (1) | (2) | (3) | (4) |
| | *TE* | *TE* | *TE* | *TE* |
| *Treat·Time·Type* | 0.021 * | 0.029 ** | 0.027 ** | 0.022 ** |
| | (0.012) | (0.014) | (0.013) | (0.011) |
| *Treat·Time* | 0.003 * | 0.001 * | 0.004 | 0.007 ** |
| | (0.002) | (0.001) | (0.007) | (0.003) |
| *Control* | *No* | *Yes* | *No* | *Yes* |
| *Cons_* | 0.360 *** | 0.474 *** | 0.371 *** | 0.468 *** |
| | (0.013) | (0.051) | (0.015) | (0.050) |
| *Province Fixed* | *Yes* | *Yes* | *Yes* | *Yes* |
| *Year Fixed* | *Yes* | *Yes* | *Yes* | *Yes* |
| *Interaction Fixed* | *Yes* | *Yes* | *Yes* | *Yes* |
| *N* | 651 | 651 | 651 | 651 |
| $R^2$ | 0.218 | 0.268 | 0.222 | 0.270 |

Note: ***, ** and * are significant at the level of 1%, 5% and 10%, respectively.

## 5. Discussion

Our main findings are that the pilot of policy-based agricultural insurance in China exacerbated agricultural fertilizer non-point source pollution, with a 4-year lasting impact. In fact, our research findings are not innovative. Many studies have already found that farmers' participation in insurance has a positive impact on fertilizer input, such as the works by Horowitz and Lichtenber [8], Zhong et al. [9] and He et al. [10]. However, it should be noted that the relationship between agricultural insurance and fertilizer input is inherently an empirical issue. Farmers in different countries and regions have different perceptions of different agricultural insurances, and their own risk preferences are also different, which leads to different types of moral hazard triggered by farmers' participation in agricultural insurance and thus different impacts on agricultural fertilizer input [17]. As such, it seems difficult for us to theoretically draw a unified conclusion on the relationship between agricultural insurance and fertilizer input. However, we need to note that almost all studies on this topic use only a small data sample to verify the effect of agricultural insurance on fertilizer inputs [8–14,35], and then the research conclusions obtained will not have such a strong warning effect, because it is difficult for us to determine the macro performance of agricultural insurance on fertilizer input in a specific country context. At the same time, the research conclusions of the small data sample itself did not have strong extrapolation, which is why our research chose a macro perspective. Especially in China's rural society, policy-based agricultural insurance is the main choice for farmers

to participate in insurance. If this insurance system is not conducive to the agricultural ecological environment, then we should really rethink this insurance from the perspective of system design. Our research conclusion merely confirms this. Therefore, compared with the small sample, our research has a stronger warning effect for system designers and has more practical significance. All in all, our research finds that the system design of policy-based agricultural insurance should take into account the negative environmental effects; otherwise, this insurance system may pose a greater threat to the agricultural ecological environment.

## 6. Conclusions

Using the panel data of 31 provinces in China (excluding Hong Kong, Macau and Taiwan) from 2000 to 2020, this paper regarded the pilot extension of policy-based agricultural insurance as a quasi-natural experiment and used the DID model to evaluate the environmental effect of policy-based agriculture insurance. The research results show that the pilot of policy-based agricultural insurance in China aggravated agricultural fertilizer non-point source pollution. After a series of robustness tests, the research conclusion was still valid. At the same time, the effect of policy-based agricultural insurance aggravating agricultural fertilizer non-point source pollution had a lasting impact for 4 years during the pilot period and did not disappear until the policy-based agricultural insurance was fully covered. In addition, the heterogeneity results show that areas in eastern China and high-disaster areas, policy-based agricultural insurance had a stronger effect on the aggravation of agricultural fertilizer non-point source pollution.

In light of the fact that policy-based agricultural insurance will aggravate agricultural fertilizer non-point source pollution, it is necessary to strengthen the monitoring of farmers' fertilization degrees and link the compensation mechanism with the degree of chemical fertilizer application in the design of insurance contracts. If farmers excessively apply chemical fertilizer, their right to enjoy corresponding income compensation should be limited. Specifically, an excess clause related to fertilizer application should be designed to increase farmers' moral hazard costs, thereby reducing farmers' excessive fertilizer application behaviors. At the same time, since farmers in eastern China and high-disaster areas are more likely to have moral hazard with overuse, these areas should be taken as the first pilot areas for the implementation of the new system to reduce the occurrence of moral hazard with overuse among farmers in these areas.

As the moral hazard problem has been rooted in the farmers themselves, the training and guidance of farmers cannot be slack. Since there is also a law of diminishing marginal returns between the fertilizer and crop yield, more fertilizer is not always better. Therefore, regular seminars or training sessions can be held to popularize agricultural knowledge among farmers, improve their sense of integrity and reduce the possibility of moral hazard. At the same time, on one hand, we should further improve the construction of the agricultural insurance market system, ensuring the transparency of the information of the agricultural insurance market. On the other hand, we should also add an institutional platform conducive to the expression of interests for the information feedback of all parties in the agricultural insurance market to reduce the information asymmetry of the agricultural insurance market, create a good operating environment for the agricultural insurance market and reduce the occurrence of moral hazard problems.

In addition, it is undeniable that there are still some research limitations in this paper. On the one hand, this paper only explored the relationship between policy-based agricultural insurance and agricultural fertilizer non-point source pollution. Whether the research conclusions of other types of agricultural insurance are consistent with this paper remains to be further tested by follow-up research. On the other hand, while being limited to the data acquisition restrictions at the macro level, this paper only controlled the main factors affecting agricultural fertilizer non-point source pollution. In order to make the research more perfect, the follow-up research should supplement relevant control factors as much

as possible on the basis of data availability to further improve the interpretation ability of the model.

**Author Contributions:** Conceptualization, Z.N. and F.Y.; methodology, Z.N.; software, Z.N; validation, Z.N., F.Y. and C.C.; formal analysis, Z.N.; investigation, F.Y. and C.C.; resources, Z.N.; data curation, Z.N. and F.Y.; writing—original draft preparation, Z.N.; writing—review and editing, Z.N. and F.Y. All authors have read and agreed to the published version of the manuscript.

**Funding:** This research received no external funding.

**Institutional Review Board Statement:** Not applicable.

**Informed Consent Statement:** Not applicable.

**Data Availability Statement:** All data generated or analyzed during this study are included in this published article.

**Conflicts of Interest:** The authors declare no conflict of interest.

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
