# Peer review of "Agricultural Insurance and Agricultural Fertilizer Non-Point Source Pollution: Evidence from China’s Policy-Based Agricultural Insurance Pilot"

_sustainability, doi:10.3390/su14052800_

Round 1

Reviewer 1 Report

The research presented is clear and comprehensive and contributes to insurance and environmental policy research.The article deals with a very current topic, which is the need to develop agricultural insurance and its impact on environmental problems related to fertilization. The topic is inspiring due to the various results obtained by other researchers of the problem. In my opinion this reserch it is  of great practical significance. This review is current  relevant and of interest to the scientific community.

The content succinctly is described and contextualized with respect to previous and present theoretical background and empirical research on the topic. 

The dfigurs / tables / diagrams are appropriate, but not include the reserch source.

The statements and conclusions are made consistent and supported by the quotes mentioned in the introduction part and the second part this reserch. But, in my opinion there is a lack of in-depth discussion of the results of empirical studies. It is advisable to introduce in the part 4 „Results and Discussions” a discussion of the results obtained by the researchers this raport with the results of the authors of similar studies.

I rate the work highly.

Author Response

Response to Reviewer 1 Comments

Review1#:

The research presented is clear and comprehensive and contributes to insurance and environmental policy research.The article deals with a very current topic, which is the need to develop agricultural insurance and its impact on environmental problems related to fertilization. The topic is inspiring due to the various results obtained by other researchers of the problem. In my opinion this reserch it is  of great practical significance. This review is current  relevant and of interest to the scientific community.

The content succinctly is described and contextualized with respect to previous and present theoretical background and empirical research on the topic. 

The dfigurs / tables / diagrams are appropriate, but not include the reserch source.

The statements and conclusions are made consistent and supported by the quotes mentioned in the introduction part and the second part this reserch. But, in my opinion there is a lack of in-depth discussion of the results of empirical studies. It is advisable to introduce in the part 4 „Results and Discussions” a discussion of the results obtained by the researchers this raport with the results of the authors of similar studies.

I rate the work highly.

Response: Thank you for this suggestion. We added the reserch source. Please see the bottom of Figure1, Table2 and Table3 in the revised manuscript. At the same time, we also strengthened the discussion of the results and the connection with relevant literature. Please see “5. Discussion” and paragraph 2 of “4.1. Estimation Results of DID Model” in the revised manuscript.

Reviewer 2 Report

The article was prepared on a topical issue, the results of the study can be used by scientists, specialists and civil servants.

It should be noted that the results of the studies are based on data that are not available in this work. Therefore, it is advisable to provide the initial data in the form of accompanying materials (tables in Excel) for a possible complete and accurate assessment of the research results.

Also, it is important to note that the latest data are from 2017, at the same time the current year is 2022.

Author Response

Response to Reviewer 2 Comments

Review2#:

The article was prepared on a topical issue, the results of the study can be used by scientists, specialists and civil servants. It should be noted that the results of the studies are based on data that are not available in this work. Therefore, it is advisable to provide the initial data in the form of accompanying materials (tables in Excel) for a possible complete and accurate assessment of the research results. Also, it is important to note that the latest data are from 2017, at the same time the current year is 2022.

Response: Thank you very much for your comments. We uploaded the data in the form of Excel as an attachment. At the same time, it is worth noting that we limit the date to 2017 for the following reasons. First, after 2017, there is a large lack of some data, which is not enough to support the research needs. Second, our research found that after the full coverage of policy-based agricultural insurance, the positive impact of policy-based agricultural insurance on agricultural fertilizer non-point source pollution will disappear, which means that limiting the time to 2017 or 2022 will be the same research findings and will not change the relevant research conclusions of this paper.

Reviewer 3 Report

  1. More details about the insurance policy should be provided; otherwise I don't really see how this policy provide motivation to the increased use of fertilizer and subsequent environmental problem. That is, how is the fertilizer use linked to the agricultural policy should be discussed further.
  2. The contribution is missing in the introduction. Please specify how this work adds knowledge to current literature.
  3. Figure 1 is not informative and the authors should redo this figure with more details.
  4. The section 2 has little about the theory and I dont see how this discussion can be related to economic or financial actions. Please elaborate.
  5. It is hard to read equations as they are in poor format. In addition, equations are spreading across the manuscript and too many definitions here and there. Calculations, equations, and variable definitions are not results, and they should not appear in the results and discussion section.
  6. The conclusion should be more concise and focal. Please remove the study description and focus more on the key findings.
  7. Move implications to the last part of discussion section because there should not have any discussion after conclusion.

Author Response

Response to Reviewer 3 Comments

Review3#:

More details about the insurance policy should be provided; otherwise I don't really see how this policy provide motivation to the increased use of fertilizer and subsequent environmental problem. That is, how is the fertilizer use linked to the agricultural policy should be discussed further.

Response: Thank you for this suggestion. We added more details to explain how agricultural insurance affects fertilizer input. Please see paragraph 1 of “2. Mechanism Analysis and Disagreement”, paragraph 2 of “2.1. The Occurrence Mechanism of Moral Hazard with Overuse” and paragraph 3 of “2.2. The Occurrence Mechanism of Moral Hazard with Nonfeasance” in the revised manuscript.

The contribution is missing in the introduction. Please specify how this work adds knowledge to current literature.

Response: Thank you for this suggestion. We add the relevant contribution of this paper in the introduction. Please see paragraph 5 of “1. Introduction in the revised manuscript.

Figure 1 is not informative and the authors should redo this figure with more details.

Response: Thank you for this suggestion. We added more details to the Figure1. Please see Figure1 in the revised manuscript.

The section 2 has little about the theory and I don’t see how this discussion can be related to economic or financial actions. Please elaborate.

Response: Thank you to the reviewers for pointing out the problems. We change the second part to “2. Mechanism Analysis and Disagreement”, which leads to our research framework by combing through the relevant literatures to explain how agricultural insurance can change fertiliser input. Please see “2. Mechanism Analysis and Disagreement in the revised manuscript.

It is hard to read equations as they are in poor format. In addition, equations are spreading across the manuscript and too many definitions here and there. Calculations, equations, and variable definitions are not results, and they should not appear in the results and discussion section.

Response: Thank you to the reviewers for pointing out the problems. We modified the equation format and moved these equations from the discussion section to the model section. Please see equation(1)-equation(5) and “3.1. Model” in the revised manuscript.

The conclusion should be more concise and focal. Please remove the study description and focus more on the key findings.

Response: Thank you for this suggestion. We removed the study description from the conclusion. Please see paragraph 1 of “6. Conclusions” in the revised manuscript.

Move implications to the last part of discussion section because there should not have any discussion after conclusion.

Response: Thank you for this suggestion. We have regrouped the conclusion section so that no discussion appears after the conclusion. Please see “6. Conclusions” in the revised manuscript.

Reviewer 4 Report

You have managed to present a very interesting work, so below there are some recommendations so as to improve further your work. So, please take them into account in your study:

A very clear introduction which allows the reader to get a first understanding of the topic and the purpose of this paper is evident. A well written introduction and ideas are presented around agricultural insurance and fertiliser in China, the policies associated to it and studies that have been conducted on the topic. However, at the end of the introduction the structure of the paper should be presented.

In the literature, the theoretical background is presented along with the framework that the study will adopt. However, any studies adopting the same or similar framework could be included to further support the choice made.

In the third part the model used is presented and the variables included are defined and well explained. In table 3, results from R2 are quite low which indicates that the independent variables are possibly not very good predictors of the dependent. Could you possibly explain why the value is low? Was that expected?

You explain well the methods used and I like a lot the way you interpret the results. However, you could point the limitations associated to the techniques used.

I would expect before the conclusion a discussion to be provided where the significance of the results will be developed and discussed with existing literature. You need to show whether you expected these results or not, based on literature, and in case these were not expected, you need to explain where the difference may lie. Thus, the significance of the results has to be shown.

Conclusion draws together the main highlights of your work and you manage to provide very nice policy implications. Please make the last sentence of the paper a bit shorter by splitting it into two sentences to improve the narration.

Although conclusion and policy implications have been provided, you need also to add the limitations of your work. In addition, further research is missing. You need to explain what else can be done on this topic and what aspect of it could be further explored or what different methodology/data can be applied.

Lastly, you have used relevant and appropriate sources however you could increase the number of references.

I hope everything makes sense!

Overall, very nice work!

Author Response

Response to Reviewer 4 Comments

Review4#:

You have managed to present a very interesting work, so below there are some recommendations so as to improve further your work. So, please take them into account in your study:

A very clear introduction which allows the reader to get a first understanding of the topic and the purpose of this paper is evident. A well written introduction and ideas are presented around agricultural insurance and fertiliser in China, the policies associated to it and studies that have been conducted on the topic. However, at the end of the introduction the structure of the paper should be presented.

Response: Thank you for this suggestion. We add the structure of the paper at the end of the introduction. Please see paragraph 6 of “1. Introduction in the revised manuscript.

In the literature, the theoretical background is presented along with the framework that the study will adopt. However, any studies adopting the same or similar framework could be included to further support the choice made.

Response: Thank you for this suggestion. We added the research expression of relevant literature in the second part. Please see “2. Mechanism Analysis and Disagreement in the revised manuscript.

In the third part the model used is presented and the variables included are defined and well explained. In table 3, results from R2 are quite low which indicates that the independent variables are possibly not very good predictors of the dependent. Could you possibly explain why the value is low? Was that expected?

Response: Thank you to the reviewers for pointing out the problems. Limited to the availability of data, the control variables we can choose are limited. Therefore, the R2 of the model is low, but this will not affect our relevant research conclusions. We regard it as a research deficiency and list it at the end of the article. We will further improve it in the follow-up research. Please see paragraph 4 of “6. Conclusions” in the revised manuscript.

You explain well the methods used and I like a lot the way you interpret the results. However, you could point the limitations associated to the techniques used.

Response: Thank you for this suggestion. We added the main limitations of the DID method. Please see paragraph 3 of “3.1. Model” in the revised manuscript.

I would expect before the conclusion a discussion to be provided where the significance of the results will be developed and discussed with existing literature. You need to show whether you expected these results or not, based on literature, and in case these were not expected, you need to explain where the difference may lie. Thus, the significance of the results has to be shown.

Response: Thank you for this suggestion. We added a discussion section. Please see “5. Discussion” in the revised manuscript.

Conclusion draws together the main highlights of your work and you manage to provide very nice policy implications. Please make the last sentence of the paper a bit shorter by splitting it into two sentences to improve the narration.

Response: Thank you for this suggestion. We divided the last sentence into two sentences. Please see paragraph 3 of “6. Conclusions” in the revised manuscript.

Although conclusion and policy implications have been provided, you need also to add the limitations of your work. In addition, further research is missing. You need to explain what else can be done on this topic and what aspect of it could be further explored or what different methodology/data can be applied.

Response: Thank you for this suggestion. At the end of the conclusion, we added the research deficiency of this paper. Please see paragraph 4 of “6. Conclusions” in the revised manuscript.

Lastly, you have used relevant and appropriate sources however you could increase the number of references.

Response: Thank you for this suggestion. We added the number of references. Please see references section in the revised manuscript.

I hope everything makes sense!

Overall, very nice work.

Round 2

Reviewer 2 Report

The authors didn't add new data to the study (last year is 2017).

Author Response

Review2#:

The authors didn't add new data to the study (last year is 2017).

Response: Thank you to the reviewers for pointing out the problems. We have updated the data for this article to 2020. Please see Table2, Table3, Table4, Table5, Table6, Table7, Figure2, and Figure3 in the revised manuscript.

Reviewer 3 Report

I am fine with this revision.

Author Response

Review3#:

I am fine with this revision.

Response: Thank you very much for the first round of comments. After being revised according to the comments, our article was improved greatly.

Reviewer 4 Report

Well done for accounting for all my comments! very nice piece of work!

Author Response

Review4#:

Well done for accounting for all my comments! very nice piece of work!

Response: Thank you very much for the first round of comments. After being revised according to the comments, our article was improved greatly.

Round 3

Reviewer 2 Report

Dear authors!

The article can be published in the current version